# Robust Multi-Scenario Speech-Based Emotion Recognition System [note 1]

**DOI:** 10.3390/s22062343

**Published:** 2022-03-18

**Authors:** Fangfang Zhu-Zhou, Roberto Gil-Pita, Joaquín García-Gómez, Manuel Rosa-Zurera

**Affiliations:** Department of Signal Theory and Communications, University of Alcalá, 28805 Alcalá de Henares, Madrid, Spain; roberto.gil@uah.es (R.G.-P.); joaquin.garciagomez@uah.es (J.G.-G.); manuel.rosa@uah.es (M.R.-Z.)

**Keywords:** affective computing, emotion recognition, speech emotions

## Abstract

Every human being experiences emotions daily, e.g., joy, sadness, fear, anger. These might be revealed through speech—words are often accompanied by our emotional states when we talk. Different acoustic emotional databases are freely available for solving the Emotional Speech Recognition (ESR) task. Unfortunately, many of them were generated under non-real-world conditions, i.e., actors played emotions, and recorded emotions were under fictitious circumstances where noise is non-existent. Another weakness in the design of emotion recognition systems is the scarcity of enough patterns in the available databases, causing generalization problems and leading to overfitting. This paper examines how different recording environmental elements impact system performance using a simple logistic regression algorithm. Specifically, we conducted experiments simulating different scenarios, using different levels of Gaussian white noise, real-world noise, and reverberation. The results from this research show a performance deterioration in all scenarios, increasing the error probability from 25.57% to 79.13% in the worst case. Additionally, a virtual enlargement method and a robust multi-scenario speech-based emotion recognition system are proposed. Our system’s average error probability of 34.57% is comparable to the best-case scenario with 31.55%. The findings support the prediction that simulated emotional speech databases do not offer sufficient closeness to real scenarios.

## 1. Introduction

Affective computing is a multidisciplinary area composed of computer science, psychology, engineering, and sociology, among many other specialties. It has gained attention of researchers, particularly with the fast advancement of new technologies, effortless access to smart devices, and the development of social networks that have made the quantity of available audiovisual information increase immeasurably in recent decades.

Moreover, speech is referred to as the most natural means of communication between humans, and not only can ideas be revealed but also speakers’ moods, attitudes, and feelings; consequently, emotions could be interpreted through non-linguistic components of speech. Furthermore, speech could be a precious information provider in human interactions because emotions cannot be effortlessly controlled or hidden.

Three primary groups of speech organs form the simplified speech production: the lungs, larynx, and vocal tract. The source-filter theory of voice production encapsulates speech as a mixture of a sound source (larynx) and a linear acoustic filter (vocal tract), which is a resonator that will customize the sound source [1]. A “power supply” is created by the lungs and airflow is offered to the larynx stage, and the larynx modulates the airflow from the lungs, triggering the vocal fold’s vibration, from which frequency provides the sound pitch. The vocal tract, defined as the connection between the larynx and the entrance—mouth and nose—obtains either a periodic puff-like or a loud airflow from the larynx and spectrally changes this input, acting as a filter.

Emotions produce unintentional physiological reactions, causing adjustments in the speaker’s speech organs and modifying the acoustic parameters [2,3,4]. For instance, fear induces an increased heart rate to send more blood to the extremities and encourage the escape response. The general muscle tension brought on by this emotion, including the face’s muscles, will most likely tense the vocal folds, turning into a raising of the fundamental frequency and modifying the subject’s voice.

Emotional speech recognition (ESR) attempts to identify and detect human emotions by capturing their voice attributes. Multiple real-world applications employ ESR, such as an in-car emotion recognition system to distinguish drivers’ moods [5] resulting in safer driving. It can certainly also be applied in call centers, specifically where the detection of callers’ emotional states might be crucial—police headquarters, emergency call centers, and medical facilities—or where corporations need to enhance the quality of service [6].

The majority of the research on ESR uses databases created under ideal laboratory conditions [7,8], i.e., real-world noise is not contemplated. Nevertheless, some studies concluded that noise is a significant factor to consider [9,10,11], seeking to develop a robust system facing noise; however, what happens when reverberation is present or when the type and level of noise change remains unclear.

In this sense, this paper pretends to study the robustness of emotion recognition systems over these two factors: noise and reverberation. Firstly, we analyze the effectiveness of emotional speech classifiers regarding the different scenarios existing in an audio recording. Although the authors have previously studied the dependency on the noise level in [12], the objective now is to extend said study to evaluate whether there is dependence on the type of noise and environment characterized by the reverb level. Thereby, in a second phase, virtual augmentation techniques are proposed to design robust classifiers, initially to the noise level and later to the type of noise and the typology of the environment.

The paper is structured as follows: Firstly, Section 2 briefly presents related works on ESR in the literature. Section 3 discusses chosen features, shortly describes overfitting and the logistic regression method, presents details of the database used and reverberation and noise effects are explained. It is followed by Section 4, where the proposed virtual augmentation of the dataset is detailed. Next, Section 5 displays outcomes, highlighting its limitations, and lastly, the paper is closed with Section 6, in which a brief discussion and the conclusion on the results is pointed out.

## 2. Related Work

In this section, related work on ESR is briefly overviewed, presenting how the field has evolved over the years. A more in-depth discussion about different approaches to address the ESR task have been developed and can be found in [13,14,15,16].

Emotion recognition might be viewed as a simple process for individuals. Nevertheless, even misjudgments are made by human beings when trying to analyze the emotions of others, with a 20.38% error probability in a study performed with four different emotions (anger, fear, happiness, and sadness) [17] or a 27.52% error probability at multimodal emotion recognition with six different emotions (anger, disgust, fear, happiness, sadness, and surprise) in a study which also includes visual content [18]. These outcomes suggest that there are difficulties in puzzling out others’ emotions. The main reasons behind it are that every speaker has their figure of speech, including a particular accent, language, pronunciation, tone, and rhythm pattern.

Furthermore, several emotions’ distinctions co-exist, varying the number of emotions among them. A collection of different classifications can be found in [19]. However, Ekman’s emotional scale [20] is frequently used in affective computing, where the fundamental emotions are declared as anger, disgust, fear, happiness, sadness, and surprise.

The first research paper about ESR [21] one can find in the literature was published in 1996. For the experiments, the paper’s authors used their own recorded corpus composed of 1000 utterances of 4 regulated emotions: anger, fear, happiness, and sadness. The study reports a 20.5% error using prosodic features related only to the fundamental pitch, comparable with the 18% of human performance error on the same recordings. The technique was based on the k-Nearest Neighbor (KNN) algorithm and a majority voting of subspace specialists technique.

After this seminal work, many other traditional approaches have been explored, in parallel with the automatic speaker recognition predecessor, using prosodic and spectral features, such as Hidden Markov Models (HMM) [22,23,24], Gaussian Mixture Models (GMM) [25,26], or Support Vector Machines (SVM) [27].

Eventually, with the increasing popularity of neural networks within all research fields, many other deep learning approaches have been investigated, such as in [28], where Recurrent Neural Networks (RNN) are applied in order to extract features as well as for classification tasks, achieving 61.8% recognition rate on the acted IEMOCAP corpus, which includes five acted emotions—anger, happiness, frustration, sadness, and neutral. Although neural networks and deep learning are undoubtedly successful for audio based applications—both as high-level feature extractors from raw data and as end-to-end classifiers—it has been shown that this approach sometimes is predisposed to present overfitting problems in the ESR design [29,30]. This overfitting is mainly caused by the lack of large datasets for audio-based emotion recognition, among other factors, due to the difficulties brought by the manual labeling and disagreements within the different annotators [31]. Thus, traditional approaches are still in use nowadays.

## 3. Materials and Methods

A brief description of the classification method is contained in this section. First, feature parameters employed for the current study are presented, and overfitting is concisely covered. Then, the choice of the classifier is briefly discussed. Finally, the effects of reverberation and noise will be briefly explained.

### 3.1. Feature Extraction

Feature parameters are initially obtained from the audio recordings. They are developed to catch the distinctive characteristics of an individual’s speech in mathematical parameters.

#### 3.1.1. Mel-Frequency Cepstral Coefficients (MFCCs)

The Mel-Frequency Cepstral Coefficients (MFCCs) are widely used as short-term acoustic features in speech and audio processing [32,33]. Because of their properties, they suit most effectively when used with classification techniques more connected to traditional machine learning, accomplishing outstanding results, for example, 85.08% of accuracy with four emotions in [34]. MFCCs are commonly calculated as follows:The audio samples are split into short overlapping segments.The signal acquired in these segments/frames is then multiplied by a Hamming window function, and the Fourier power spectrum is obtained.A non-linear Mel-space filter-bank analysis is carried out, and the logarithm is then calculated.-The filter-bank analysis generates the spectrum energy in every channel—also known as the filter-bank energy coefficients—representing separate frequency bands.-The logarithm operation extends the scale of the coefficients and also decomposes multiplicative elements to additive.Lastly, MFCCs are obtained by performing a Discrete Cosine Transform (DCT) on the filter-bank log-energy parameters and maintaining several leading coefficients. DCT has two fundamental properties:It compacts the energy of a signal into a few coefficients.Its coefficients are extremely decorrelated. This attribute helps the models that consider feature coefficients to be uncorrelated.

In summary, the power spectrum, logarithm, and DCT sequence of operations produce the well-known cepstral representation.

MFCCs represent the spectral envelope of the speech signal, therefore acquiring pertinent characteristics of speech. For example, the first MFCC coefficient offers information about signal energy, and the second MFCC coefficient points out the energy balance between low and high frequencies.

#### 3.1.2. Delta Mel-Frequency Cepstral Coefficients (ΔMFCCs) and Delta-Delta Mel-Frequency Cepstral Coefficients (ΔΔMFCCs)

Even though MFCCs describe the stationary characteristics of each frame, the dynamic attributes are not obtained. Temporal evolution could be represented by calculating the first (ΔMFCCs) and second derivatives (ΔΔMFCCs) of cepstral coefficients. The former, known as velocity coefficients, represent the variation of the MFCC coefficients over a time instant. Similarly, the latter are called acceleration coefficients considering that they return the variation of the ΔMFCC coefficients over a time instant.

#### 3.1.3. Pitch

It is a perceptual parameter that determines the perceived tone frequency. When a sound is produced, it provides information about the vibration of the vocal cords. Previous studies have demonstrated that the pitch is related to the underlying emotion [35,36].

Since this paper explores the effect of noise on ESR, the algorithm used for the pitch estimation is the one proposed in [37], where the authors demonstrate that it is an algorithm robust to high levels of noise. The algorithm merges a comb-filter used in the log-frequency power spectral domain with a non-linear amplitude compression.

### 3.2. Generalization Problems: Overfitting

In the machine learning field, overfitting is the effect of learning algorithms to become specialized only on the data they employ during the training process, causing a generalization problem, and pruned to have poor performance when new data arrive at them for being tested.

The presence of generalization and overfitting problems is mainly conditioned by two factors. First, the amount of available data is directly related to the presence of generalization problems. In general, when the data with which the model is fed are too small, overfitting is more likely to occur.

The second factor is related to the intelligence of the classifiers, that is, the capability of the classifier to implement complex solutions. Typically, the more intelligent the classifier, the more usual the presence of generalization problems. In this point, it is important to take into account that the intelligence of the classifier does not only depend on the number of weights and units that compose its structure but also on the number of input features.

Typically, many techniques are applied to detect and prevent overfitting problems, including early-stopping during training, and different types of cross-validation techniques to prevent overfitting over the test dataset (holdout method, k-fold cross-validation, or leave-one-out cross-validation). A method named bootstrapping is also used to estimate the model performance [38]. It involves iteratively choosing a particular number of arbitrary training-test subsets to evaluate different smaller databases. This strategy allows to verify the robustness and generalization of the results.

Bootstrapping method can be adjusted for the specific task of ESR, by tailoring the training-test subsets to choose specific individuals for the training set and the rest for the test set. In this way, the performance obtained by the classifiers is speaker independent, given that the test set patterns of a given subject are not part of the training sets.

### 3.3. Logistic Regression for Multiclass Classification

Machine learning took logistic regression from the statistics area as a classification algorithm. It is founded on probability [39] and assigns observations to a discrete set of classes. Its name originates from the hypothesis function that the algorithm uses, the logistic (sigmoid) function.

While some algorithms are designed for binary classification problems, such as logistic regression, they could be converted into a multiclass classifier. There are two strategies: using binary classifiers for multiclass problems or multiclass classifiers, including Naive Bayes, decision trees, and multiclass SVM. Within the first approach, the concept is to break down the multiclass problem into a collection of binary problems, generate binary classifiers for these problems, and then integrate the output of binary classifiers as a multiclass classifier [40].

The logistic regression algorithm is one of the simplest methods that exhibits better generalization capabilities, that is, ability to keep the performance under changes in the characteristics of the test dataset. As already stated, this paper aims not to improve the performance of state-of-the-art systems to solve the speech-based emotion recognition task but to explore the robustness of the classifiers under different scenarios, including changes in the noise and reverberation conditions. Therefore, the logistic regression algorithm is chosen due to its simplicity and, thus, its generalization capability. The idea is that if this unsophisticated method cannot generalize to environment conditions different from those used in the design set, then more complex solutions, such as those based on deep learning, will also suffer from the same dependence of the performance on the noise and other environmental elements.

### 3.4. Environment Effects on Recordings

Several conditions must be taken into account when designing and testing ESR systems, including codification, noise, and reverberation.

In the previous work presented by the authors in [12], codification of the speech signal was demonstrated to affect the performance of the classifiers. An emotion recognition system designed using audio extracted from a PCM signal does not keep its performance when a GSM codification is used. Fortunately, the type of codification used is a parameter known when implementing an emotion recognition system (will be related to the source of the audio files) and, therefore, its effects over the results can be minimized by simply having two different emotion recognition systems, one designed to work with PCM audios and another trained using GSM compressed audios. This fact reduces the importance of the codification over the design and thus it has not been considered in the present study.

The second factor to take into consideration is the noise characteristics. The level and type of noise in the recordings strongly depend on the environment and on the distance from the microphone to the subject. The level of noise present in the audio, related to the signal-to-noise ratio (SNR), is very important in order to design a recognition system. In [12], we already demonstrated that a classifier trained under ideally clean recordings does not keep its performance when a considerable level Gaussian white noise is introduced in the audio files. However, the type of noise and not only its level might also condition the performance of the recognition system. An ESR system trained using white Gaussian noise might lose its performance in other noisy environments such as cafeterias or living rooms, in which we cannot consider the noise to be white (equally distributed in all frequencies) or stationary (stable energy over time). Thus, these two factors (noise level and type of noise) must be taken into account, either when analyzing the performance of the recognition systems under changing environments or when designing robust systems.

The third and last factor related to the environment to consider is the reverberation. Reverberation is produced when a sound source stops emitting and the original wave produced by the source is reflected against the surrounded obstacles, such as the walls of the room or trees in a forest. This acoustic effect is known as a notable side effect when solving audio signal processing problems, such as automatic speech recognition or the one that concerns the present paper, e.g., ESR. Reverberation can be simulated by convolving a given audio signal *x*—which represents the sound source—and the Room Impulse Response (RIR), *h*: y=x∗h. The RIR will depend on the size of the room, the presence of obstacles, the reflection coefficient of the materials, the position of the microphone, and the position of the speaker. It will also depend on the orientation of the microphone and the speaker in those cases in which their spatial response is not omnidirectional. Since reverberation has already been demonstrated to affect the performance of a multitude of audio signal processing algorithms (e.g., separation, localization, and classification algorithms), we will study the dependency of the performance of an ESR system under different reverberation conditions.

## 4. Proposed Virtual Enlargement through Noise and Reverberation Addition

Virtual enlargement consists of virtually increasing the size of the design set by generating new synthetic patterns. As already discussed in previous sections, one of the main problems in ESR is the small size of the available databases. As stated above, increasing the size of the training set tends to reduce the generalization and overfitting problems and can be very beneficial when only small datasets are available—which is usually the case when working with ESR. In this sense, virtual enlargement and data augmentation has already been used to improve the performance of emotion recognition systems in the literature. In [41], a method for generating synthetic files by modifying the speed and the pitch of the audio files was used to improve the performance of speech emotional classifiers.

In a similar way, in this paper, we propose a method for generating new instances of the database by synthetically modifying the environmental conditions of the recordings. The proposed strategy for addressing this problem is a virtual enlargement simulating different scenarios in which different types of noise and reverberation can be synthetically added to the original files.

The main idea is to replicate the same patterns of a database with different SNRs several times and then stack them together. Given a database, the parameters for the proposed virtual enlargement are:Number of design patterns, *N*.Minimum signal-to-noise ratio, $SNR_{min}$: the desired minimum value of signal-to-noise ratio in decibels (dB) to add to each audio file of the database.Maximum signal-to-noise ratio, $SNR_{max}$: the desired maximum value of signal-to-noise ratio in decibels (dB) to add to each audio file of the database.Step factor, $\Delta_{SNR}$: an integer value that indicates the increment between $SNR_{min}$ and $SNR_{max}$ to take into account for the virtual enlargement. Starting from $SNR_{min}$—included—$\Delta_{SNR}$ will take regularly spaced values until it reaches $SNR_{max}$.

The number of replicated databases is:(1)$$M_{r}\;=\;\left\lfloor \frac{SNR_{max}-SNR_{min}+1}{\Delta_{SNR}}\right\rfloor$$

In addition, the total number of patterns of the virtual enlarged database will be:(2)$$N_{noise}\;=\;N\cdot M_{r}$$

For instance, a virtual enlargement with $SNR_{min}$ = 1 dB and $SNR_{max}$ = 40 dB, and step factor $\Delta_{SNR}$ = 8 dB, the training set is created by stacking the datasets database +8 dB,

database +16 dB, database +24 dB, database +32 dB, and database +40 dB of the corresponding scenario. Figure 1 shows this example.

Furthermore, if we not only consider a type of noise but $M_{n}$ different noises (for instance, Gaussian noise, cafeteria, noise, music, etc.) and $N_{r}$ different reverberating scenarios, we can further increase the size of the design patterns.
(3)$$N_{total}\;=\;N\cdot M_{r}\cdot M_{n}\cdot M_{r}$$

It is important to highlight that, in this paper, the objective of this enlargement is not to improve the performance for a given scenario but to generate more robust classifiers, able to suitably perform, independently of the characteristics of the scenario: noise level, noise typology, and reverberation level. In this sense, the next section pretends to explore the beneficial effects that the proposed data augmentation technique might have under changes in the conditions of the environment.

## 5. Results

The experimental setup is first presented within this section, followed by the experiments’ results and the constraints that ought to be considered.

### 5.1. Experimental Setup

The Berlin Database of Emotional Speech [42] has been chosen for this paper as it has been widely used in other proposals, such as in [8,43,44,45,46], and it is considered a classic database for ESR. The original database encompasses 535 audio records produced by 10 actors, 5 females and 5 males. The utterances are generated in the German language, tagged with seven different emotions, anger, boredom, disgust, anxiety/fear, happiness, sadness, and neutral, which are a subset of Ekman’s model [20]. For every utterance executed by an actor, there could be different variations. Seven repeated audios with the same utterance created by the same actor have been eliminated to avoid material redundancy. A summary of the database information can be found in Table 1.

As specified in previous sections, the edited database used to perform the experiments includes 528 audio files produced by 10 actors—5 females and 5 males—referred to as the “base database” to differentiate it from the original database with 535 audio files. As a result, the dataset is not large enough to prevent the system from overfitting. The strategy proposed for addressing this problem is a virtual enlargement described in Section 4 of the base database. In order to analyze the effect of different scenarios, another four main datasets are created considering:White Gaussian Noise Scenario (WGNS).Real-World Noise Scenario (RWNS).Reverberated White Gaussian Noise Scenario (RWGNS).Reverberated Real-World Noise Scenario (RRWNS).

In the case of Real-World Noise, the MUSAN dataset [47] is employed, which sums up 42 h and 31 min of music from several genres, 60 h of speech from 12 different languages, and approximately 6 hours of technical and non-technical noises. The addition is performed by randomly picking an audio file from the entire MUSAN dataset without replacement.

Furthermore, in the reverberating scenarios, i.e., RWGNS and RRWNS, reverberation is performed by convolving Room Impulse Response (RIR) proposed by McGovern in [48] with reflection coefficient of 0.3, room dimensions of 20 m × 19 m × 21 m, coordinates of the sound source (5, 2, 1), and coordinates of the microphone (19, 18, 1.6). With these parameters, the reverberation time is measured, obtaining RT${}_{60}=0.5$ s.

#### 5.1.1. Creation of Test Sets

For each of the previously mentioned scenarios, another 41 databases are generated based on the original 528 audio files, adding the corresponding type of noise ranging from 40 dB signal-to-noise ratio—representing a nearly clean speech from a human hearing capability perspective—to 0 dB, which involves the incorporation of a considerable noisy level. Figure 2 shows an example of the followed procedure for one of the four scenarios.

#### 5.1.2. Creation of Training Sets

For each type of scenario, the procedure described in Section 4 is followed using the virtual enlargement parameters $SNR_{min}$ = 1 dB and $SNR_{max}$ = 40 dB. Twenty different training sets/models are created with different $\Delta_{SNR}$ values ranging from $\Delta_{SNR}$ = 1 to $\Delta_{SNR}$ = 20. For each of these 20 models, all 41 test sets presented above are tested, and then the average error probability is calculated.

#### 5.1.3. Feature Extraction

Once the databases are created, the following features are extracted from the raw audios: MFCCs, two features originated from the latter, ΔMFCCs and ΔΔMFCCs, and the Pitch. Two statistics (mean and standard deviation) are employed to acquire the final features. The features used in the conducted experiments are summarized in Table 2.

#### 5.1.4. Classifier and Evaluation Details

The classifier is set up with L2-regularized logistic regression with a one-vs.-all scheme and 100 maximum iterations for the solver to converge. Moreover, audio normalization is performed to the raw audio files, and feature scaling is applied.

For a model evaluation, many training and test set subdivisions are made using bootstrapping: one female and one male are selected as the test set for each iteration in this study. The rest of the individuals, e.g., 4 females and 4 males, are left as the training set, performing 25 different classification tasks. The error probability is calculated for each iteration. After the 25 iterations, the average error probability is calculated.

#### 5.1.5. Time Sequence of Experiments

The experiments are implemented in three main blocks. First, the initial stage is performed by making use of simple databases—without virtual enlargement—with a signal-to-noise ratio of 20 dB. Different scenarios are cross tested, from which results have shown that various noises and reverberation influence performance. Due to this, a second stage is conducted. The experiments on this second stage seek to identify the number of databases gathered in the training set to stabilize the test set error. Finally, after distinguishing the minimum number of patterns needed for significant performance, a robust speech-based emotion recognition system is developed against the different scenarios, i.e., different types and levels of noises, and reverberation.

### 5.2. Numerical Results

The first stage of experiments shows worse results when a classifier is tested with the same patterns but adding a signal-to-noise ratio of 20 dB of different types of noises, and vice versa. These results are shown in Table 3 in which the best performance is achieved in the best-case scenario, i.e., when the training set and test set are coming from the same distribution, represented in bold by the diagonal of the table.

Regarding the second stage of experiments, Figure 3, Figure 4, Figure 5 and Figure 6 display the average error probability for each model, trained only with the training set relating to the same scenario, i.e., WGNS, RWNS, RWGNS, and RRWNS, respectively. Each figure shows error probabilities as a percentage on the Y-axis against the dataset tested as the signal-to-noise ratio added to the base dataset in dB from 0 to 40 in increments of 1 on the X-axis. Three lines are shown corresponding to the models with $\Delta_{SNR}$ = 1 dB, $\Delta_{SNR}$ = 3 dB, and $\Delta_{SNR}$ = 20 dB, and each line graph contains the datasets marked with a dot, i.e., the training set is constituted by those test set databases indicated by the dots. From these figures, the following conclusions could be gathered:The model $\Delta_{SNR}$ = 20 dB has the most unstable error probability in all four cases. As it reveals, virtually enlarging the dataset helps stabilize the error probability throughout all test sets.White Gaussian Noise scenarios, both reverberated and non-reverberated (Figure 3 and Figure 5), have a larger gap between the model $\Delta_{SNR}$ = 20 dB (green line) than $\Delta_{SNR}$ = 1 dB and $\Delta_{SNR}$ = 3 dB when comparing against Real-World scenarios.Real-World Noise scenarios, both reverberated and non-reverberated (Figure 4 and Figure 6), tend to be more sensitive on noisier test sets when comparing with the White Gaussian Noise scenarios, where we can see more stable results for noisier test sets.Real-World Noise Scenario (Figure 4) seems to be the only case where for cleaner test sets it does not show an improvement with a smaller virtual enlargement, i.e., $\Delta_{SNR}$ = 20 dB, to even larger training set models.

Table 4 shows a summary of the previously presented results from Figure 3, Figure 4, Figure 5 and Figure 6, including the twenty different models—step factors $\Delta_{SNR}$ from 1 to 20 dB. A single-tailed z-test is performed to study the significance level of these results, where:The null hypothesis is that the error probability resulting from a given model is equal or lower, i.e., better, than the probability resulting from using $\Delta_{SNR}$ = 1dB, where value of 1 dB is selected as the comparison score, since it performs the best average error probabilities as it contains the largest number of patterns available for the virtual enlargement.The alternative hypothesis is that the error probability of the model $\Delta_{SNR}$ = 1 dB is lower than the one resulting from another given model, i.e., it represents that a change has occurred compared to the situation described by the null hypothesis.

Table 5 shows p-values using a significance level α of 0.05 from the different results provided by Table 4. As can be seen, the alternative hypotheses are rejected in those cases in which $\Delta_{SNR}>3$dB in at least one of the four main databases. This implies that there might not be a loss of probability between training with the model $\Delta_{SNR}$ = 1 dB and $\Delta_{SNR}$ = 3 dB —the null hypothesis cannot be rejected—, and hence there should not be meaningful differences when comparing training with 21,120 and 7920 patterns. Thus, we have selected $\Delta_{SNR}$ = 3 dB for the next batch of experiments in which we will look for combining multiple types of noises and environments in the enlargement of the training dataset so that a robust classifier is generated.

Finally, the same databases as the first stage (Table 3) are tested making use of virtual enlargement with $\Delta_{SNR}$ = 3 dB with five different models: each scenario isolated and a combination of all four scenarios. Table 6 shows a comparison, where the first column represents the best-case scenario extracted from Table 3. As can be seen, although the best-case scenario, where the test and training sets are obtained from the same database, shows a better error probability, it could be noticed that the performance on the “all scenarios” is comparable with the former results, making it a robust model.

Furthermore, same experiments as explained before are performed with the CREMA-d corpus [49], which contains 7442 clips of 91 actors containing 6 emotions—happiness, sadness, anger, fear, disgust and neutral—crowd sourcing labeled from 2443 raters. Table 7 shows its results and, as can be seen, although the performances are worst than the ones provided at Table 6, it still supports our findings presented before. Note that the error probability deterioration is caused by the fact that the CREMA-d corpus is harder to classify since EMO-db corpus audio files have better quality than the former.

Additionally, Figure 7 shows the normalized confusion matrix for the proposed model “All scenarios $\Delta_{SNR}$ = 3 dB”, tested with “Base 40 dB” on the EMO-db corpus.

As can be seen, the best well-classified emotion corresponds to sadness with 92% of the cases. Additionally, a notable error rises when classifying happiness, confused with anger on 32% of the cases, and also anxiety/fear emotion is misinterpreted as fear.

## 6. Discussion and Conclusions

Many studies concerning the detection of emotions by analyzing speech are performed under laboratory conditions. This implies that the employed audios from databases for the experiments are under synthetic conditions: emotional states are pretend and the environment does not have noise, but those are fictional scenarios.

This paper analyzes the side effects of different noisy and reverberating environments in emotion classification systems. Experiments are performed with a signal-to-noise ratio of 20 dB, which is an insignificant change to the hearing. However, the results reveal that the level and the type of noise do affect the developed systems as a result of a poor generalization of the input data: performance decreases from 25.57% to 79.13% of error probability in the worst case.

The results stated above prompted a more extensive analysis. The database used for this paper is virtually enlarged, with various signal-to-noise ratios to generate a more robust system by stacking them as a more extensive training set. Outcomes reveal that regardless of the well-known evidence that the larger the training set is, the better the performance becomes, there are no relevant differences between training with 21,120 patterns ($\Delta_{SNR}$ = 1 dB) and 7,392 patterns ($\Delta_{SNR}$ = 3 dB), although it does stabilize the system results.

Furthermore, a robust multi-scenario speech-based emotion recognition system is developed comparable to the best-case scenario—where the designer knows in advance the type and level of noise of audios to be tested—by stacking different databases from different scenarios with the minimum number of patterns required for the results to be significant.

Considering that the objective of this article is not to improve the performance of state-of-the-art systems but to investigate the impact of noise from different scenarios, neither advanced algorithms nor complex features have been utilized. Moreover, languages of the databases used in this paper bias the results. Future implementations might employ other algorithms and features and use emotional databases containing different languages.

## Figures and Tables

**Figure 1 sensors-22-02343-f001:**
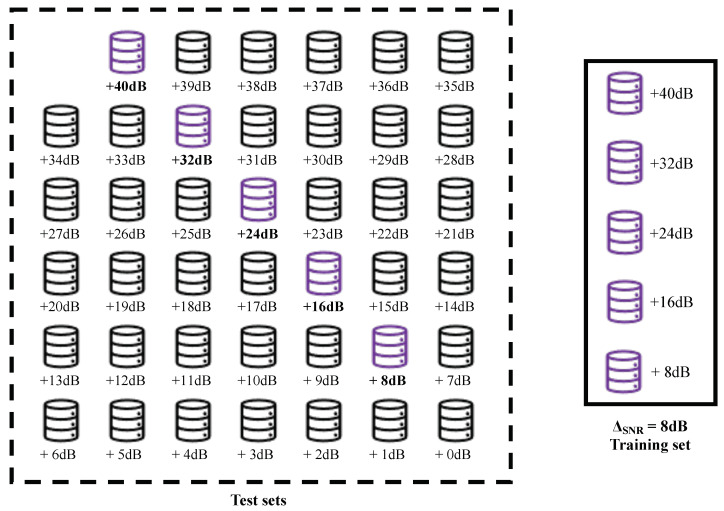
Diagram of the creation of virtual enlarged training set for one scenario using $SNR_{min}$ = 1 dB and $SNR_{max}$ = 40 dB and stepping factor $\Delta_{SNR}$ = 8 dB.

**Figure 2 sensors-22-02343-f002:**
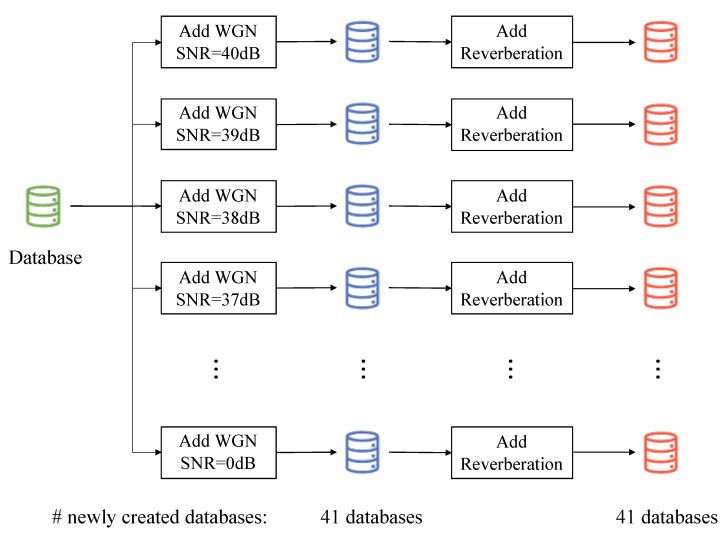
Creation of test sets diagram for one scenario.

**Figure 3 sensors-22-02343-f003:**
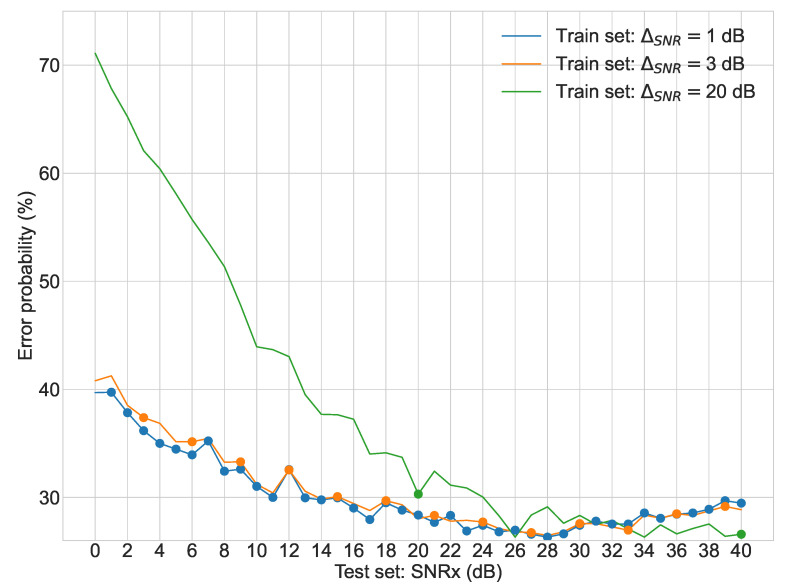
Second stage experiment results: test sets error probabilities for three different models (White Gaussian Noise Scenario).

**Figure 4 sensors-22-02343-f004:**
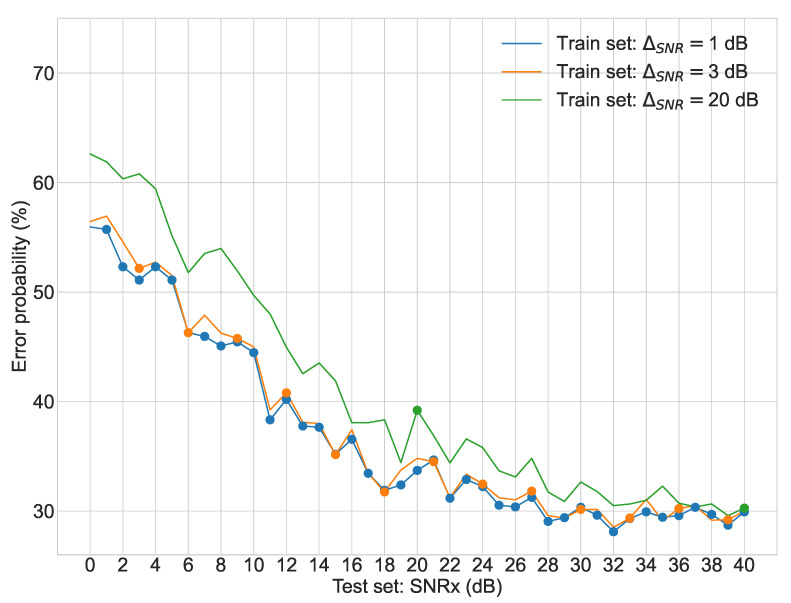
Second stage experiment results: test sets error probabilities for three different models (Real-World Noise Scenario).

**Figure 5 sensors-22-02343-f005:**
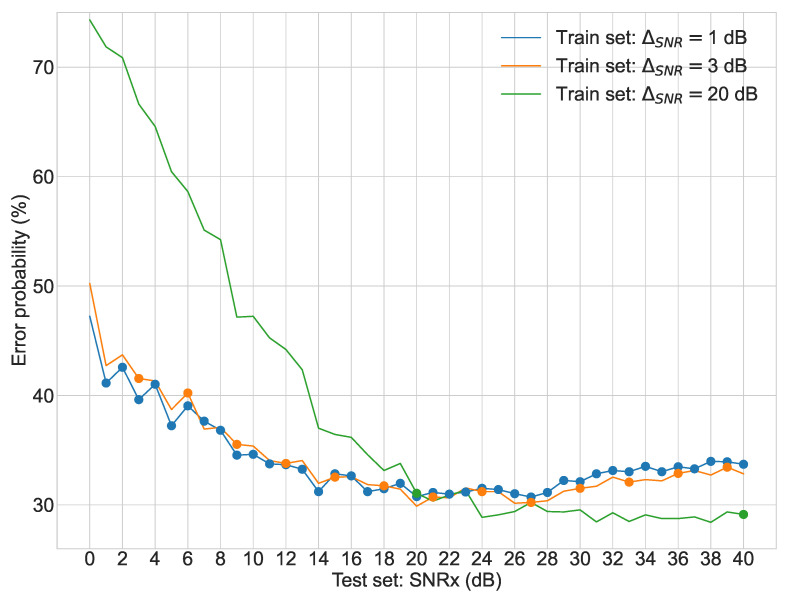
Second stage experiment results: test sets error probabilities for three different models (Reverberated White Gaussian Noise Scenario).

**Figure 6 sensors-22-02343-f006:**
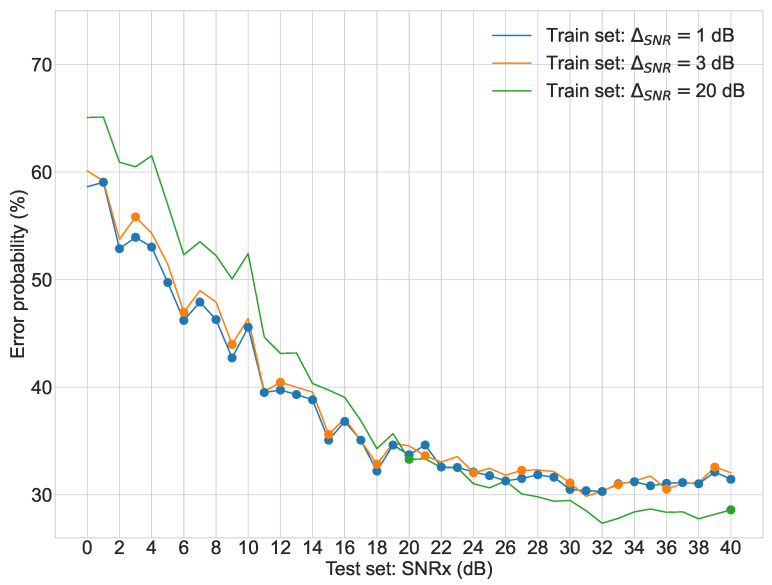
Second stage experiment results: test sets error probabilities for three different models (Reverberated Real-World Noise Scenario).

**Figure 7 sensors-22-02343-f007:**
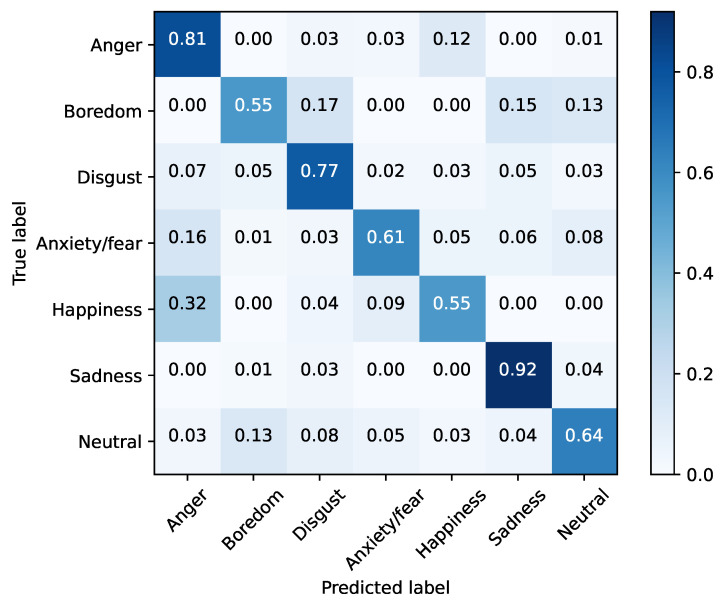
Normalized confusion matrix of the original database as test set and the proposed model “All scenarios $\Delta_{SNR}$ = 3 dB” as training set on the EMO-db corpus.

**Table 1 sensors-22-02343-t001:** Emotion distribution of the edited Berlin Database used.

Emotion	Number of Audios Files
Anger	126
Boredom	80
Disgust	46
Anxiety/fear	68
Happiness	69
Sadness	62
Neutral	77
Total Number of Audio Files	528

**Table 2 sensors-22-02343-t002:** Extracted features for the experiments.

Statistic	Feature	Index	Total Number of Features
Mean	MFCCs (13 coef.)	1–13	40
	ΔMFCCs (13 coef.)	14–26	
	ΔΔMFCCs (13 coef.)	27–39	
	Pitch	40	
Standard Deviation	MFCCs (13 coef.)	41–53	40
	ΔMFCCs (13 coef.)	54–66	
	ΔΔMFCCs (13 coef.)	67–79	
	Pitch	80	

**Table 3 sensors-22-02343-t003:** First stage experiment results. Error probability (%) cross results within different noises databases.

	Classifier					
Test Set	Base 40 dB	WGNS 20 dB	RWNS 20 dB	Reverb.	RWGNS	RRWNS
Base 40 dB	**25.57**	72.80	32.80	77.84	78.75	68.79
WGNS 20 dB	63.71	**30.42**	42.92	46.17	53.90	51.93
RWNS 20 dB	55.49	71.06	**40.08**	74.36	79.96	69.13
Reverb.	64.62	66.86	56.70	**27.50**	55.04	38.11
RWGNS 20 dB	78.90	72.27	56.55	62.73	**30.49**	44.28
RRWNS 20 dB	79.13	68.45	59.77	51.97	50.38	**35.23**

**Table 4 sensors-22-02343-t004:** Second stage experiment results. Average error probability (%).

Training Set		Average Error Probability (%)			
**Step Factor** $\Delta_{\mathrm{SNR}}$ **(dB)**	**Total Number of** **Design Patterns**	**WGNS**	**RWNS**	**RWGNS**	**RRWNS**
1	21,120	30.24	36.81	34.19	37.85
2	10,560	30.27	37.12	34.28	37.62
3	7920	30.54	37.32	34.20	38.40
4	5280	30.58	37.58	34.33	37.96
5	4224	30.88	38.53	34.15	39.06
6	3168	31.16	37.80	34.82	38.56
7	2640	31.75	38.57	34.17	39.30
8	2640	31.76	38.96	35.00	38.81
9	2112	32.47	39.52	34.84	39.66
10	2112	32.45	39.68	35.00	39.18
11	1584	33.11	38.56	35.24	39.08
12	1584	33.05	39.05	35.87	39.13
13	1584	33.87	39.80	36.56	39.22
14	1056	34.31	41.04	37.27	40.06
15	1056	35.08	41.47	37.62	39.86
16	1056	35.56	40.99	38.47	39.89
17	1056	36.37	41.83	39.72	40.43
18	1056	37.29	41.22	38.68	39.49
19	1056	38.72	41.03	39.85	40.08
20	1056	38.32	40.94	39.81	39.34

**Table 5 sensors-22-02343-t005:** P-value for some model results from Table 4.

Training Set	*p*-Value			
**Step Factor** $\Delta_{\mathrm{SNR}}$ **(dB)**	**WGNS**	**RWNS**	**RWGNS**	**RRWNS**
2	0.461	0.187	0.401	0.751
3	0.199	0.070	0.492	0.053
4	0.170	0.013	0.347	0.375
5	0.038	0.000	0.548	0.000
6	0.005	0.002	0.034	0.018

**Table 6 sensors-22-02343-t006:** Third stage experiments results. Error probability (%) cross results within different noises databases (Emo-DB).

	Classifier					
	Best Case	WGNS	RWNS	RWGNS	RRWNS	All Scenarios
Test Set	Scenario	$\Delta_{\mathrm{SNR}}$ = 3 dB	$\Delta_{\mathrm{SNR}}$ = 3 dB	$\Delta_{\mathrm{SNR}}$ = 3 dB	$\Delta_{\mathrm{SNR}}$ = 3 dB	$\Delta_{\mathrm{SNR}}$ = 3 dB
Base 40 dB	25.57	30.49	29.13	56.89	56.59	30.34
WGNS 20 dB	30.42	28.03	39.09	47.65	45.38	30.15
RWNS 20 dB	40.08	48.26	34.81	62.27	59.17	36.63
Reverb.	27.50	63.18	59.66	37.27	37.16	40.64
RWGNS 20 dB	30.49	64.28	63.67	29.89	38.14	33.71
RRWNS 20 dB	35.23	72.42	61.17	40.76	34.51	35.95
Average	**31.55**	51.11	47.92	45.79	45.16	**34.57**

**Table 7 sensors-22-02343-t007:** Third stage experiments results. Error probability (%) cross results within different scenarios (CREMA-d).

	Classifier					
	Best Case	WGNS	RWNS	RWGNS	RRWNS	All Scenarios
Test Set	Scenario	$\Delta_{\mathrm{SNR}}$ = 3 dB	$\Delta_{\mathrm{SNR}}$ = 3 dB	$\Delta_{\mathrm{SNR}}$ = 3 dB	$\Delta_{\mathrm{SNR}}$ = 3 dB	$\Delta_{\mathrm{SNR}}$ = 3 dB
Base 40 dB	49.67	50.71	52.11	74.12	72.36	51.97
WGNS 20 dB	49.36	50.49	54.19	74.32	71.88	52.33
RWNS 20 dB	54.11	57.58	53.54	74.45	72.59	53.82
Reverb.	49.97	66.09	61.30	51.68	53.27	54.94
RWGNS 20 dB	49.56	67.66	65.71	50.36	52.48	53.87
RRWNS 20 dB	54.27	71.42	64.93	55.90	53.55	56.07
Average	**51.16**	60.66	58.63	63.47	62.69	**53.83**

## Data Availability

The Emo-DB database used in this paper is publicly available at http://emodb.bilderbar.info/download/, accessed on 29 April 2021.

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
