# Peer review of "Robust Multi-Scenario Speech-Based Emotion Recognition Systemâ€"

_sensors, 2022, doi:10.3390/s22062343_

Round 1
Reviewer 1 Report
The paper is well-written. The problem which is discussed is very significant for emotion recognition in real situations, not done in laboratory conditions.
I recommend separating related work from the introduction.
Also, English should be checked (e.g. in line 285 "has been chosen" is repeated twice). Moreover, the present time should be used when sections are described in the introduction.
Reviewer 2 Report
Advantages:
The whole language of the article is concise, with clear structure, and can better discuss the innovation points.
Disadvantages:
- Only one corpus was used for the experiment, with fewer samples.
2.The origin of the 20 models mentioned in the 270 lines of the article is not explained.
- Rows 383 to 387 in the article explain why 3dB will be chosen next, but it is too rough to make it difficult for readers to understand.
